

# A dark past, a restrained present, and an apocalyptic future: time perspective, personality, and life satisfaction among anorexia nervosa patients

Danilo Garcia[1,2,3,4], Alexandre Granjard[1], Suzanna Lundblad[5] and Trevor Archer[2,3]

[1] Blekinge Centre of Competence, Blekinge County Council, Karlskrona, Sweden
[2] Department of Psychology, University of Gothenburg, Gothenburg, Sweden
[3] Network for Empowerment and Well-Being, Sweden
[4] Department of Psychology, Lund University, Lund, Sweden
[5] Psychiatry Affective, Anorexia & Bulimia Clinic for Adults, Sahlgrenska University Hospital, Gothenburg, Sweden

Corresponding author
Danilo Garcia,
danilo.garcia@icloud.com

## ABSTRACT

**Background.** Despite reporting low levels of well-being, anorexia nervosa patients express temperament traits (e.g., extraversion and persistence) necessary for high levels of life satisfaction. Nevertheless, among individuals without eating disorders, a balanced organization of the flow of time, influences life satisfaction beyond temperamental dispositions. A balanced time perspective is defined as: high past positive, low past negative, high present hedonistic, low present fatalistic, and high future. We investigated differences in time perspective dimensions, personality traits, and life satisfaction between anorexia nervosa patients and matched controls. We also investigated if the personality traits and the outlook on time associated to positive levels of life satisfaction among controls also predicted anorexia patients' life satisfaction. Additionally, we investigated if time perspective dimensions predicted life satisfaction beyond personality traits among both patients and controls.

**Method.** A total of 88 anorexia nervosa patients from a clinic in the West of Sweden and 111 gender-age matched controls from a university in the West of Sweden participated in the Study. All participants responded to the Zimbardo Time Perspective Inventory, the Ten Item Personality Inventory, and the Temporal Satisfaction with Life Scale.

**Results.** A $t$-test showed that patients scored higher in the past negative, the present fatalistic, and the future dimensions, lower in the past positive and the present hedonistic dimensions, higher in conscientiousness, extraversion, and agreeableness, and lower in life satisfaction. Regression analyses showed that life satisfaction was predicted by openness to experience and emotional stability for controls and by emotional stability among patients. When time dimensions were entered in the regression, emotional stability and the past negative and past positive time dimensions predicted life satisfaction among controls, but only the past positive and present hedonistic time dimensions predicted life satisfaction among patients.

**Conclusion.** Anorexia patients were less satisfied with life despite being more conscientious, social, and agreeable than controls. Moreover, compared to controls, patients had an unbalanced time perspective: a dark view of the past (i.e., high past negative), a restrained present (i.e., low present hedonistic) and an apocalyptic view of the future

(i.e., high present fatalistic). It is plausible to suggest that, therapeutic interventions should focus on empowering patients to cultivate a sentimental and positive view of the past (i.e., high past positive) and the desire to experience pleasure without concern for future consequences (i.e., high present hedonistic) so that they can make self-directed and flexible choices for their own well-being. Such interventions might have effects on life satisfaction beyond the patients' temperamental disposition.

## INTRODUCTION

*"[. . .] and in addition to the feeling of being full there was another more terrifying one, as if a hundred appetites were raging out of control within her. She couldn't explain it, but she felt as if everything was in chaos and something awful was going to happen. She had eaten and now something terrible would occur."*

*In 'The Best Little Girl in the World' by Steven Levenkron (Levenkron, 1978).*

Eating disorders present a serious and potentially life-threatening condition. This is especially true for anorexia nervosa, which is characterized by episodes of self-starvation, the preference for being underweight, a distorted body image, and an intense fear of gaining weight (*Lee, Cloninger & Chaea, 2015*). This line of research shows that diet and eating behaviors are strongly associated with particular personality characteristics in clinical samples (e.g., *Lee, Cloninger & Chaea, 2015*; *Anckarsäter et al., 2012*), making personality a major factor of eating disorders in general (*Lilenfeld et al., 2006*; *Cassin & Von Ranson, 2005*). Eating disorder patients describe themselves as anxiety prone, with low self-esteem, and poor emotional self-regulation. Besides these self-descriptions, anorexia nervosa patients also describe themselves as rigid, industrious/persistent, and perfectionists (e.g., *Akan & Grilo, 1995*). These personality profile and their propensity to experience anxiety and negative affect (*Harney et al., 2014*; *Spring & Bulik, 2014*; *Lundblad et al., 2015a*; *Lundblad et al., 2015b*) might lead to cognitive-emotional dysfunctions, such as, intolerance of uncertainty (cf. *Sternheim, Startup & Schmidt, 2011*; *Sternheim et al., 2011*; *Sternheim, Startup & Schmidt, 2015*). This behavior is almost synonymous with neuroticism, but at the same time these patients present a picture of the self as conscientious and persistent. Paradoxically, conscientiousness and persistence have been linked to well-being and happiness (*Garcia, 2011a*; *Garcia, 2012*; *Garcia et al., 2013*), but eating disorder patients, especially anorexia nervosa patients, express an absence of positive emotions and lack of satisfaction in several planes of their lives (*Kitsantas, Gilligan & Kamata, 2003*; *Phillipou et al., 2015*; *Dapelo et al., 2016*; *Brand-Gothelf et al., 2014*; *Lavender et al., 2016*; *Tomba et al., 2014*). Thus, the question is if, as it is among individuals without eating disorders, these personality traits are predictors of life satisfaction among anorexia nervosa patients or if there are other aspects of the being that are associated with life satisfaction across healthy and unhealthy individuals.

In this context, the Science of Well-being suggests that individual differences in personality, influence the physical, psychological, and social aspects of health and well-being of all individuals (*Cloninger, 2004*; *Garcia & Rosenberg, 2016*). Nevertheless, instead of linking specific temperament traits, such as extraversion and neuroticism, to well-being, Cloninger's Science of Well-Being focuses on self-awareness of the unity of being through sustainable personality development in three dimensions (*Cloninger, 2004*; *Cloninger, 2006*; *Cloninger, 2007*; *Cloninger, 2013*; *Garcia & Rosenberg, 2016*). In Cloninger's paradigm, unity of being represents a complete, coherent and harmonious sense of the self, giving raise to feelings of hope in one self's ability to cope and make decisions (i.e., Self-directedness), feelings of love and empathy towards others (i.e., Cooperativeness), and feelings of faith in that we are part of something beyond the self and others (i.e., Self-transcendence). In short, the ability to be self-aware and organize one's life in a coherent manner is more important to well-being than specific temperament traits, such as, extraversion and persistence.

Indeed, in line with their personality characteristics (i.e., high levels of neuroticism and persistence), anorexia nervosa patients display an inability or difficulty to evaluate the future long-term outcomes of their actions/choices and also an incapacity or inflexibility to adapt their mindset/strategies to new information or novel environmental stimuli (*Tenconi et al., 2016*). Thus, perhaps explaining their intolerance of uncertainty (cf. *Sternheim et al., 2011*; *Sternheim, Startup & Schmidt, 2011*). The cognitive ability to organize the past, present, and future (i.e., a person's time perspective) in a coherent manner is indeed an important aspect of an individual who can make self-directed choices for her/his own well-being and flexible adapt to new situations (*Cloninger, 2004*; *Garcia et al., 2016*). As a matter of fact, a balanced organization of the flow of time seems to influence life satisfaction beyond temperament traits (*Zhang & Howell, 2011*).

A balanced or coherent time perspective may be described as: a sentimental and positive view of the past (i.e., high past positive), a less pessimistic attitude toward the past (i.e., low past negative), the desire to experience pleasure without concern for future consequences (i.e., high present hedonistic), a less fatalistic and hopeless view of the future (i.e., low present fatalistic), and the ability to find reward in achieving specific long-term goals (i.e., high future) (*Zimbardo & Boyd, 1999*; *Zimbardo, Keough & Boyd, 1997*). Individuals' perspective of time predicts the reported use of alcohol, drug, and tobacco (*Keough, Zimbardo & Boyd, 1999*), risky driving (*Zimbardo, Keough & Boyd, 1997*; *Boyd & Zimbardo, 2005*), indecision and avoidant procrastinations (*Díaz-Morales, Ferrari & Cohen, 2008*), environmental engagement (*Milfont, Wilson & Diniz, 2012*), the choice of food and of partner, educational achievement, and the distinctness of future goals (*Zimbardo & Boyd, 1999*). Time perspective is also related to well-being–negative views of the past have been reported to be associated with depression, whereas positive views of the past have been reported to be associated with happiness and life satisfaction (*Bruehlman-Senecal & Ayduk, 2015*; *Zimbardo & Boyd, 1999*; see also *Stolarski et al., 2014*; *Sailer et al., 2014*; *Garcia et al., 2016*). In relation to anorexia patients, however, if positive information of the present, or past for that matter (e.g., they have become thinner), is not well adjusted into their mindset (e.g., "I'm still overweight"), then even if long-term goals are future oriented, these goals might not be healthy in nature (e.g., "I need to eat less").

## The present study

To the best of our knowledge, no other study has examined if anorexia nervosa patients actually differ in their outlook of time from controls. Interesting, the few studies on eating disorders suggest that individuals who are preoccupied with immediate concerns and pleasures and who are unable to comprehend how their current actions will affect them in the future (i.e., high in the present hedonistic and low in the future time perspectives) may be more likely to place themselves at risk for binge eating and binge drinking (*Boyd & Zimbardo, 2005*). Probably due to the considerable direct positive outcomes (i.e., pleasure and disinhibition) linked to binge behaviors (*Laghi et al., 2012*). Additionally, individuals who engage in binge behaviors also present an outlook oriented towards a fatalistic present, which might cloud the sense of control they have over their actions and also influence them to lack hope for the future (*Laghi et al., 2012*). It is plausible that anorexia nervosa patients also have this specific outlook on time, along a time perspective associated with low life satisfaction and psychological distress (i.e., high past negative, low past positive, high present fatalistic) (cf. *Culbert et al., 2016*; *Seidel et al., 2016*). After all, anorexia nervosa patients share other forms of risky behaviours, low levels of life satisfaction, depression, and anxiety with patients with eating disorders in general. However, we expected that anorexia nervosa patients in comparison to controls would delay gratification by restraining themselves from engaging in pleasant behaviors in the present and to focus on negative consequences of their current actions (i.e., less present hedonistic), and also by visualizing and formulating future goal states that shape their current judgments and decisions (i.e., high future oriented). That is, anorexia patients are expected to express high levels of future time perspective and conscientiousness (i.e., industrious and perfectionists), but at the same time they are also expected to express high levels of neuroticism and an incoherent time perspective (i.e., high past negative, low past positive, high present fatalistic) and low life satisfaction.

In addition to differences in personality, time perspective, and life satisfaction between anorexia nervosa patients and controls, we also investigated the effect of personality and outlook on time on individuals' level of life satisfaction. Specifically, we investigated if the personality traits and the outlook on time associated to positive levels of life satisfaction among controls, also predicted anorexia patients' life satisfaction. Additionally, we investigated if time perspective dimensions predicted life satisfaction beyond personality traits. We found this important in order to shade light on other aspects of the being, in this case the ability to organize the flow of time, that might influence well-being beyond temperament traits. If it is so, this will give suggestions for the development or use of interventions that target well-being rather than ill-being or symptoms. After all, the absence of life satisfaction and positive emotions is a serious problem because it is more predictive of subsequent mortality and morbidity than the presence of negative emotions (*Huppert & Whittington, 2003*). For instance, although some reviews of meta-analyses show that cognitive behavioral therapy is effective for different disorders (e.g., unipolar depression, generalized anxiety disorder, panic disorder with or without agoraphobia, social phobia, posttraumatic stress disorder, and childhood depressive and anxiety disorders) (*Butler et al., 2006*), single studies and other meta-analyses show either only moderate positive effects on mental health (e.g., *Wang et al., 2007*) or mixed results (*Hofmann et al., 2012*;
*Moore et al., 2016*). Moreover, specific reviews of the anorexia nervosa literature suggest that no single psychological intervention has clear superiority in treating these patients, thus, highlighting the importance of the development of novel interventions (*Bodell & Keel, 2010*). In this context, therapeutic interventions targeting the development of life satisfaction, sense of autonomy and control, goal-directed behavior, self-empowerment, cooperativeness, empathy, and sense of commitment and belonging to society (i.e., well-being therapy or person-centered well-being coaching) have been shown to provide alleviation for mental health problems and disabilities, such as affective and behavioral disorders, in general and clinical populations (*Cloninger, 2004*; *Cloninger, 2006*; *Fava et al., 2005*; *Fava et al., 1998*; *Fava & Ruini, 2003*; *Fava & Tomba, 2009*; *Ruini et al., 2006*; *Ruini et al., 2014*; *Ruini, Albieri & Vescovelli, 2014*; *Ruini et al., 2017*). Randomized controlled trials of well-being therapy, in patients with mental disorders, show improvements, such as, treatment adherence, less relapse and recurrence rates when compared with cognitive behavioral therapy or psychotropic medications alone (*Cloninger, 2006*).

## METHODS

### Ethical statement

The data at the clinic was collected as part of the development of technics and also as a follow up of the therapies conducted at the clinic. After consultation with the central ethical review board we arrived at the conclusion that the design of the study (e.g., all participants' data were anonymous to the researchers and will not be used for commercial or other non-scientific purposes) required only informed consent from the participants.

### Participants and procedure

Patients ($N = 88$ females, age range between 24 to 45 years) presenting either restrictive anorexia nervosa or anorexia nervosa with both starving combined with binge eating and purging participated in the study. All anorexia patients had a history of unsuccessful interventions and were referred from the Department of General Psychiatry, Sahlgrenska University Hospital (Gothenburg, Sweden). The healthy volunteers (i.e., 114 Controls matched by age, gender, education, socioeconomic status) were university students selected from a larger sample collected for another study (*Sailer et al., 2014*). The patient group had all been afflicted with the eating disorder symptoms for over five years on arrival at the Anorexia and Bulimia Clinic for adults (Sahlgrenska University Hospital). To the best of our knowledge, the patients did not suffer from severe malnutrition when they participated in the study.

All the patients who were contacted agreed to participate. On arrival at the clinic, each patient was met by the respective professional workers, nurse, psychologist, or physician and then completed the Eating Disorder Inventory-2 to measure symptoms of eating disorder and was given their diagnosis by the staff. Thereafter, patients received instructions and were allowed 30–45 min to respond to the battery of instruments. The university students in the control group were asked to participate during class hours and were allowed the same amount of time for completing the survey.

## Measures

### Time perspective

The Zimbardo Time Perspective Inventory (*Zimbardo & Boyd, 1999*) consists of 56 items that measure the following five time dimensions: Past Positive (e.g., "It gives me pleasure to think about my past"), Past Negative (e.g., "I think about the good things that I have missed out on in my life"), Present Hedonistic (e.g., "Taking risks keeps my life from becoming boring"), Present Fatalistic (e.g., "Fate determines much in my life"), and Future (e.g., "I believe that a person's day should be planned ahead each morning"). The Swedish version has been used in previous studies and showed good psychometric properties (e.g., *Sailer et al., 2014*; *Garcia et al., 2016*; *Garcia, Nima & Lindskär, 2016*) and its psychometric properties have been validated in many different languages (*Milfont, Wilson & Diniz, 2012*; *Liniauskaite & Kairys, 2009*; *Díaz-Morales, 2006*).

### Personality

The Ten Item Personality Inventory (*Gosling, Rentfrow & Swann Jr, 2003*) is a short ten-item measure that asks to which extent (1 = *not at all*; 5 = *very much*) the participant sees herself/himself as open to experience, conscientious, extrovert, agreeable, and neurotic. The inventory measures each Big Five trait through two items for each trait (e.g., I see myself as "extraverted, enthusiastic" and "quite, reserved" as measures of extraversion).

### Temporal life satisfaction

The Temporal Satisfaction With Life Scale (*Pavot, Diener & Suh, 1998*) comprises 15-items rated on a 7-point Likert scale (1 = strongly disagree, 7 = strongly agree) assessing past (e.g., "If I had my past to live over, I would change nothing"), present (e.g., "I would change nothing about my current life"), and future life satisfaction (e.g., "There will be nothing that I will want to change about my future"). The Swedish version of the instrument has been used in previous studies (*Garcia, 2011b*; *Sailer et al., 2014*; *Garcia, Rosenberg & Siddiqui, 2011*; *Garcia, Nima & Lindskär, 2016*).

## Statistical methods

We conducted a $t$-test for independent samples in order to test differences perspective dimensions, personality, and life satisfaction between anorexia nervosa patients and matched controls. Two separated regression analyses (i.e., one for the patient sample and one for the control sample) were conducted to investigate if the personality traits and the outlook on time associated to positive levels of life satisfaction among controls, also predicted anorexia patients' life satisfaction. In both regressions, personality and time perspective dimensions were the predictors and life satisfaction the outcome. Additionally, these regressions aimed to investigated if time perspective dimensions predicted life satisfaction beyond personality traits, thus, personality was the predictor in the first model and both personality and time perspective dimensions the predictors in the second model.

## RESULTS

We first conducted a $t$-test to compare if anorexia patients differed in the five time perspective dimensions, the Big Five traits, and temporal life satisfaction. In short, anorexia
**Table 1** Means, standard deviations (Sd.), and results of the $t$-tests comparing the five time perspective dimensions, the Big Five personality traits, and temporal satisfaction with life between anorexia nervosa patients and controls.

| | Anorexia patients | | Controls | | $t$-tests' results | | |
| --- | --- | --- | --- | --- | --- | --- | --- |
| | *Mean* | *Sd.* | *Mean* | *Sd.* | *t* | *df* | *p* |
| Past negative | 3.50 | 0.75 | 2.98 | 0.80 | 4.59 | 192 | <.001 |
| Past positive | 2.78 | 0.73 | 3.44 | 0.70 | −6.46 | 195 | <.001 |
| Present fatalistic | 2.51 | 0.55 | 2.28 | 0.61 | 2.68 | 192 | <.01 |
| Present hedonistic | 2.65 | 0.65 | 3.18 | 0.55 | −6.15 | 194 | <.001 |
| Future | 3.58 | 0.52 | 3.33 | 0.57 | 3.06 | 192 | <.01 |
| Openness to experience | 4.79 | 1.46 | 4.28 | 2.45 | 1.68 | 192 | .096 |
| Conscientiousness | 4.98 | 1.53 | 3.67 | 2.36 | 4.44 | 191 | <.001 |
| Extraversion | 4.74 | 1.52 | 3.73 | 2.47 | 3.32 | 192 | <.01 |
| Agreeableness | 5.11 | 1.13 | 4.01 | 2.48 | 3.77 | 191 | <.001 |
| Emotional stability | 3.16 | 1.46 | 3.71 | 2.42 | −1.85 | 193 | .066 |
| Temporal satisfaction with life | 2.82 | 1.03 | 4.10 | 1.16 | −8.06 | 194 | <.001 |

Notes.
Pink cells mark variables in which anorexia nervosa patients scored significantly higher compared to controls; Blue cells mark variables in which controls scored significantly higher compared to anorexia nervosa patients.

patients scored higher in the past negative, the present fatalistic, and the future dimensions and higher in conscientiousness, extraversion, and agreeableness. Controls scored higher in both the past positive and the present hedonistic dimensions and also higher in temporal satisfaction with life (see Table 1).

Next, we conducted multiple regression analyses in which temporal satisfaction with life was the outcome variable and personality was the predictor in the first model and time perspective the predictor in the second model. These analyses were conducted separately for each participant group in order to investigate if the personality traits and the outlook on time associated to positive levels of life satisfaction among controls, also predicted anorexia patients' life satisfaction. Additionally, these regressions aimed to investigated if time perspective dimensions predicted life satisfaction beyond personality traits.

For anorexia patients, the multiple regression revealed that at step one, personality contributed significantly to the regression model ($F(5, 68) = 3.69$, $p < .01$) and accounted for 23% of the variation in temporal satisfaction with life. In this first step, emotional stability (positively) was the only significant predictor of temporal satisfaction with life among anorexia patients (see Table 2). Introducing the time perspective dimensions explained an additional 23% of variation in temporal satisfaction with life. This change in $R^2$ was significant ($F(10, 68) = 4.82$, $p < .001$). In this second step, emotional stability was no longer a significant predictor of temporal satisfaction with life. Instead, both the past positive (positively) and the present hedonistic (positively) time perspective dimensions predicted significantly temporal satisfaction with life among anorexia patients (see Table 2).

For participants in the control group, the multiple regression revealed that at step one, personality contributed significantly to the regression model ($F(5, 98) = 5.12$, $p < .001$) and accounted for 21% of the variation in temporal satisfaction with life. In this first step, openness to experience (negatively) and emotional stability (positively) were the significant
**Table 2** Results of the hierarchical multiple regression analyses, for each group, in which temporal satisfaction with life was the outcome, the Big Five personality traits the predictors in the first step, and the five time perspective dimensions the predictors in the second step of the regression.

| Group | Variable | β | t | R | $R^2$ | $\Delta R^2$ | p |
|---|---|---|---|---|---|---|---|
| | STEP 1 | | | .48 | .23 | .23 | |
| | Openness to experience | 0.02 | 0.15 | | | | .882 |
| | Conscientiousness | 0.18 | 1.48 | | | | .145 |
| | Extraversion | 0.22 | 1.70 | | | | .093 |
| | Agreeableness | −0.13 | −1.04 | | | | .302 |
| | **Emotional stability** | **0.33** | **2.66** | | | | **< .05** |
| | STEP 2 | | | .67 | .45 | .23 | |
| | Openness to experience | −0.10 | −0.92 | | | | .363 |
| Anorexia patients | Conscientiousness | 0.25 | 1.83 | | | | .073 |
| | Extraversion | 0.04 | 0.30 | | | | .764 |
| | Agreeableness | −0.01 | −0.08 | | | | .934 |
| | Emotional stability | 0.08 | 0.62 | | | | .539 |
| | Past negative | −0.22 | −1.86 | | | | .068 |
| | **Past positive** | **0.24** | **2.33** | | | | **< .01** |
| | Present fatalistic | −0.17 | −1.42 | | | | .161 |
| | **Present hedonistic** | **0.50** | **3.47** | | | | **< .001** |
| | Future | 0.16 | 1.14 | | | | .259 |
| | STEP 1 | | | .46 | .21 | .21 | |
| | **Openness to experience** | **−0.87** | **−3.47** | | | | **< .001** |
| | Conscientiousness | 0.05 | 0.24 | | | | .814 |
| | Extraversion | 0.36 | 1.82 | | | | .071 |
| | Agreeableness | −0.05 | −0.17 | | | | .866 |
| | **Emotional stability** | **0.76** | **3.96** | | | | **< .001** |
| | STEP 2 | | | .82 | .67 | .46 | |
| | Openness to experience | −0.06 | −0.32 | | | | .746 |
| | Conscientiousness | −0.14 | −0.83 | | | | .409 |
| Controls | Extraversion | −0.05 | −0.35 | | | | .729 |
| | Agreeableness | −0.07 | −0.36 | | | | .721 |
| | **Emotional stability** | **0.32** | **2.27** | | | | **< .05** |
| | **Past negative** | **−0.59** | **−7.86** | | | | **< .001** |
| | **Past positive** | **0.26** | **3.79** | | | | **<.001** |
| | Present fatalistic | −0.13 | −1.94 | | | | .056 |
| | Present hedonistic | 0.06 | 0.85 | | | | .397 |
| | Future | 0.10 | 1.36 | | | | .178 |

Notes.
Significant results in bold type. Pink marks results for anorexia nervosa patients and blue cells mark results for controls.

predictors of temporal satisfaction with life among controls (see Table 2). Introducing the time perspective dimensions explained an additional 46% of variation in temporal satisfaction with life. This change in $R^2$ was significant ($F(10, 93) = 18.47$, $p < .001$). In this second step, emotional stability (positively) and both the past negative (negatively)

and past positive (positively) time perspective dimensions predicted significantly temporal satisfaction with life among controls (see Table 2).

## DISCUSSION

We investigated differences in time perspective dimensions, personality traits, and life satisfaction between anorexia nervosa patients and matched controls. We also investigated if the personality traits and the outlook on time associated to positive levels of life satisfaction among controls, also predicted anorexia patients' life satisfaction. Additionally, we investigated if time perspective dimensions predicted life satisfaction beyond personality traits among both patients and controls. Our results showed that anorexia patients had a darker and less sentimental view of the past (i.e., high past negative and low past positive), a higher reluctance to experience pleasure (i.e., low present hedonistic), a more fatalistic and hopeless view of the future (i.e., high present fatalistic), and at the same time a higher tendency to find reward in achieving specific long-term goals (i.e., high future). Moreover, the patients were higher in conscientiousness, extraversion, agreeableness, and lower in life satisfaction. In other words, patients' time-awareness is placed in a dark past, an extremely restrained present, and an apocalyptic future, while their self-awareness is focused on them as hard-workers (i.e., conscientiousness), external positive stimuli (i.e., extroversion), and pleasing and being kind to other people (i.e., agreeableness). In other words, presenting a personality usually associated with happiness and well-being, but an unbalanced time perspective or an unhappy outlook on time. In essence, the personality traits measured here seem to be necessary for high levels of life satisfaction but not sufficient (cf. *Cloninger, 2004*). Instead, individuals level of self-awareness is probably what helps individuals to adapt to internal and external conditions (*Cloninger, 2004*; *Ruini, 2017*).

This picture of the self in time (i.e., time perspective) and space (i.e., personality) that the patients present goes in line with earlier studies in which anorexia patients describe themselves as rigid, industrious, and perfectionists (e.g., *Akan & Grilo, 1995*). In the present study, however, our results also suggest that these time-space characteristics might lead to cognitive-emotional-social dysfunctions in anorexia nervosa patients (cf. *Sternheim et al., 2011*; *Sternheim, Startup & Schmidt, 2011*; *Cardi et al., 2015*). For instance, a focus of the self in relation to external positive stimuli (i.e., high extraversion) in conjunction with a extremely restrictive present (i.e., low present hedonistic) and focus on future reward (i.e., high future) might lead to a persistent attitude towards self-starving in an already highly conscientious individual with an apocalyptic view of the future (i.e., high present fatalistic). This description suggests a persistent striving for future reward from and individual without hope for the future which might explain why anorexia nervosa patients display an inability or difficulty to evaluate the future long-term outcomes of their actions/choices and also an incapacity or inflexibility to adapt their mindset/strategies to new information or novel environmental stimuli (cf. *Tenconi et al., 2016*).

Unsurprisingly, for patients, a better view of the past and a more hedonistic present (i.e., a less restrictive attitude towards the present) seems to increase their life satisfaction. When it comes to the controls, the past positive dimension and the past negative dimensions

together with emotional stability were the main predictors. Suggesting that the cultivation of a sentimental and positive view of the past (i.e., high past positive) would lead to increases in life satisfaction for both patients and individuals without eating disorders, but that increases in the desire to experience pleasure without concern for future consequences (i.e., high present hedonistic) would lead to increases in life satisfaction only among anorexia nervosa patients. Moreover, personality did not seem to play the same kind of role for patients as for controls. This is against most studies on life satisfaction, which show that emotional stability or low neuroticism, along extraversion, is one of the major predictors of life satisfaction (*Pavot, Diener & Suh, 1998*). Again, temperament traits might help to define the type of disorder, but not if a person would develop a disorder (*Cloninger, 2004*). In this study, not only did patients present what the literature would partially define as a happy personality, but also these traits were not even associated with patients' life satisfaction when time perspective was introduced as a predictor. Hence, we argue that temperament traits do not define who the happy people are and that focusing on the cultivation of specific time dimension would lead to higher well-being among patients. This effect seems to go beyond the patients' temperamental dispositions. That being said, although we stated that emotionally stability played a role for controls' life satisfaction but not for patients, we cannot rule out that this difference was rather because the relationship between personality traits-life satisfaction was less strong for patients to begin with.

## Limitations and suggestions for future studies

One limitation in the present study was that we used a very short instrument to measure personality. Also in this line, although the big five is the most well-established model of personality in psychology, in psychiatry, Cloninger's biopsychosocial model (*Cloninger, 2004*) presents another way to think about personality. This model, does not only cover temperament or people's emotional reactions, but also their character or goals and values as a ternary structure of human self-awareness: self-directedness or the person's relation to herself, cooperativeness or the person's relation to others, and self-transcendence or the person's relation the world and universe as a whole. In other words, Cloninger's biopsychosocial model of personality present a system of descriptions of both pathological functioning, normal functioning and optimal functioning (*Ruini, 2017*). This personality model is a non-linear model that seems to capture the complex dynamics of personality and well-being because it takes into account the interaction within temperament traits and character traits and also the person's temperament and character profiles, that is, within and between automatic emotional reactions and systems of self-regulation (*Cloninger & Garcia, 2015*; *Garcia et al., 2017a*; *Garcia et al., 2017b*; *Garcia et al., 2017c*). By seeing personality as holographic and ternary in nature, Cloninger's model may allow a better understanding of how self-awareness in time (i.e., past, present, and future) and space (i.e., self, others, and the world as a whole) influences anorexia patients' satisfaction with life.

In addition, although matching the participants is a recommendable technique to compare, in this case, time perspective, personality and life satisfaction among patients and non-patients, it is important to point out the risk of overmatching. According to the literature, age and gender used here as confounders, are associated with the prevalence

of anorexia nervosa (*Lundblad, Hansson & Archer, 2014*; *Lundblad et al., 2015a*; *Lundblad et al., 2015b*), which might cause statistical bias (*Rubin, 1973*; *Anderson, Kish & Cornell, 1980*). Importantly, although we matched for age, the choice of university students as matched controls can be challenged on whether they are representative for the population from which the patients derived. Future studies should therefore use other variables, such as socioeconomic status, as confounders and use other populations as controls.

## Conclusion and final remarks

Anorexia patients were less satisfied with life despite being more conscientious, extrovert, and agreeable than controls. Paradoxically, these personality traits are often addressed as important for a happy life. Nevertheless, patients had an unbalanced time perspective: a dark view of the past (i.e., high past negative), a restrained present (i.e., low present hedonistic) and an apocalyptic view of the future (i.e., high present fatalistic). The patient group's dark attitude towards their history is well in line with experiences in clinical settings. Many of the patients have traumatic experiences from the past and fatalistic views of both the present and the future. Additionally, the patients' view of themselves as kind and pleasant might be demanding and make life difficult for someone with a dark apprehension of the past and the future: "people are kind and I'm agreeable but there is no hope for the future, hence no meaning". It is plausible to suggest that, therapeutic interventions should focus on empowering patients to feel safe to explore the inherently unpredictable world and on becoming more flexible or resilient (*Harrison et al., 2016*). Additionally, encouraging patients to cultivate a sentimental and positive view of the past (i.e., high past positive) and the desire to experience pleasure without concern for future consequences (i.e., high present hedonistic). For instance, interventions targeting the development of life satisfaction and character have been shown to provide alleviation for mental health problems and disabilities, such as affective and behavioral disorders, in general and clinical populations (e.g., *Cloninger, 2004*; *Cloninger, 2006*; *Fava et al., 2005*; *Ruini, 2017*; *Garcia et al., 2016*).

### Funding

The development of this article was funded by a grant from the Swedish Research Council (Dnr. 2015-01229). The funders had no role in study design, data collection and analysis, decision to publish, or preparation of the manuscript.

### Grant Disclosures

The following grant information was disclosed by the authors:
Swedish Research Council: Dnr. 2015-01229.

### Competing Interests

The authors declare there are no competing interests. Danilo Garcia is the Director of the Blekinge Center of Competence, which is the Blekinge County Council's research and development unit. The Center works on innovations in public health and practice through interdisciplinary scientific research, community projects, and the dissemination

of knowledge in order to increase the quality of life of the habitants of the county of Blekinge, Sweden.

## Author Contributions

- Danilo Garcia conceived and designed the experiments, analyzed the data, wrote the paper, prepared figures and/or tables, reviewed drafts of the paper.
- Alexandre Granjard wrote the paper, reviewed drafts of the paper.
- Suzanna Lundblad performed the experiments, reviewed drafts of the paper.
- Trevor Archer conceived and designed the experiments, wrote the paper, reviewed drafts of the paper.

## Human Ethics

The following information was supplied relating to ethical approvals (i.e., approving body and any reference numbers):

The data at the clinic was collected as part of the development of technics and also as a follow up of the therapies conducted at the clinic. After consultation with the central ethical review board we arrived at the conclusion that the design of the study (e.g., all participants' data were anonymous to the researchers and will not be used for commercial or other non-scientific purposes) required only informed consent from the participants.

## Data Availability

The raw data has been supplied as a Supplementary File.

## Supplemental Information

Supplemental information for this article can be found online at http://dx.doi.org/10.7717/peerj.3801#supplemental-information.

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
