# Peer review of "A dark past, a restrained present, and an apocalyptic future: time perspective, personality, and life satisfaction among anorexia nervosa patients"

_PeerJ, doi:10.7717/peerj.3801_

## Round 0.1 · original submission · Major Revisions

· Academic Editor

Major Revisions

My opinion is that comments and suggestions of both reviewers are well founded.

However, I am sure that you are able to reply concerns of reviewers and submit a revised version of this interesting ms. If you choose to revise, please reply to each item of both reviewers.

·

Basic reporting

The quality of writing is currently unacceptable. There are multiple errors and missing references (e.g. Sternheim et al., 2011 and Cloninger, 2004), and the authors do not cope with the “curse of knowledge” (see Pinker’s famous writing tips). It seems that almost no attention has been given for the Discussion. Much more work should go into thinking clearly and writing clearly, and would not hurt to consult a language reviewer and a statistician either. Number of properties of the sample and the measures go undiscussed. For example, all the patients “had a history of unsuccessful interventions”. Is this an important factor in the sense that it might make them different from other anorexia nervosa patients? For another example, unreliability and state-dependence of human memory has been extensively studied. The authors could better explain the Zimbardo’s time perspectives and what is known about how they relate to objective time, personality, etc. Currently, they do poor job in covering the relevant literature; for example, missing is a study titled “Do time perspectives predict unique variance in life satisfaction beyond personality traits?” (http://dx.doi.org/10.1016/j.paid.2011.02.021).

Regarding the journal's specific data requirements, I am not sure how raw the "raw data" should be. The authors have provided the aggregate variables used in the analyses, but not the wider set of variables from which they were aggregated (i.e., no raw data on numeric answers of the patients and controls). Perhaps this is sufficient.

Experimental design

The authors have collected a reasonably well-powered sample of anorexia nervosa patients and matched controls, and find multiple differences between the groups. Design of analyses could be reconsidered, however. The first analysis (Table 1) characterizes the univariate differences between anorexia patients and controls with respect to the studied variables, and makes perfect sense to me. However, I wonder if the second analysis (Table 2) could be replaced or supplemented with something more informative than “what explains life satisfaction within the study groups”, such as “what explains the difference in life satisfaction between the study groups”. I assume that few are interested in making anorexics happy anorexics, while many are interested in normalizing their psyche. In practice, this could amount e.g. to studying a regression model where group membership (anorexic=1, control=0) is a predictor of life satisfaction, and examining which ‘adjusting’ covariates attenuate its regression coefficient the most. It would, of course, also be interesting if something could be said about time perspectives and objective treatment outcomes.

Validity of the findings

The reported degrees of freedom in lines 199 and 209 give away that the authors have been testing significance of the total model instead of the change in R-square, which they write about. Furthermore, the Table 2 should preferably give confidence intervals for the coefficients, and authors should address possible multicollinearity e.g. by reporting variance inflation factors. The reader would probably also wish to see a correlation matrix for the studied variables as descriptive data. If multicollinearity is a problem (an unlikely case), authors could consider remedies such as employing Elastic Net regression, or simply analyze separately the variables causing the multicollinearity. In addition, it is advisable to avoid the term “hierarchical”, because this gets confounded with multilevel modeling and stepwise regression techniques. Better to simply refer to multiple regression with different blocks, or sets, of variables.

Additional comments

The authors of the paper #15827, titled “Dark past, a restrained present, and an apocalyptic future: time perspective, personality, and life satisfaction among anorexia nervosa patients”, posit that Zimbardo’s time perspectives have not been previously studied in anorexic patients. I am not well-versed with this literature, but it appears that the setting is a fresh one. The authors have collected a reasonably well-powered sample of anorexia nervosa patients and matched controls, and find multiple differences between the groups. While the study setting appears scientifically relevant and interesting, the manuscript has number of weaknesses that can be addressed, but may require a lot more work (please, see the journal's review sections 1-3).

Reviewer 2 ·

Basic reporting

The paper is well written and interesting. However, the paper could be clearer if some comments were addressed:

1) Introduction: “The Science of Well-being” might be unknown to readers – please explain in more detail.

2) Please include a paragraph of statistical methods. More details are needed to understand.

3) Please acknowledge study´s limitations more clearly in discussion.

4) The study’s argument remains bit vague:

A) Discussion paragraph 2: This is very interesting. Please discuss in more detail using easy to understand language. Overall, put your main findings in context.
B) In the abstract the authors’ state: “patients’ outlook of contemplation of the past and pleasure in the present might have implications for well-being interventions…” In the conclusion paragraph, could you explain how your findings could influence the treatment of anorexia nervosa? Should clinicians, psychotherapists be aware or address some things based on your findings?

Experimental design

1) The selection of controls in case-control study can introduce bias. In this case it seems that chosen controls differs from the cases in other respects than eating disorder status. The authors should justify matching in more detail and mention all the relevant information throughout the paper.

2) Can overmatching introduce bias in the paper? These details should also be discussed in the limitation section of the paper.

3) In the models multicollinearity and overfitting are concern. Please explain.

Validity of the findings

As previously mentioned.

Additional comments

1) The authors state that all anorexia nervosa patients had a history of unsuccessful interventions and were referred from the Department of General Psychiatry. Did patients suffer from severe malnutrition when they participated the study?

2) Are there any previous studies how self-evaluation of personality changes in anorexia nervosa patients when they get better/gain normal weight? Could you discuss this in more detail, if it is relevant?

---

## Round 0.2 · Minor Revisions

· Academic Editor

Minor Revisions

Dear Danilo,
Please take into account the reviewer's comments.

·

Basic reporting

The authors have been responsive to my previous comments and do much better job at explaining the role of “time perspective” in anorexia than before, although this has not resolved all problems with the text. For example, a citation to “Cassin & van Ranson, 2005” has been added to the first page (tracked changes), but cannot be found from the reference list. Straw man argumentation should be removed. For example, the authors sell their topic writing: “For instance, interventions based on cognitive behavioral therapy show only moderate positive effects on mental health (e.g., Wang, Simon, Avorn, Azocar, Ludman et al., 2007).” But this is just a one study on a different disorder, using a sampling process not comparable to theirs, whereas there are tens of meta-analyses on CBTs, mostly reporting large effect sizes (https://doi.org/10.1016/j.cpr.2005.07.003). In addition, parts of the text are too vague, like the conclusion in the abstract: “It is plausible to suggest that, therapeutic interventions should focus on empowering patients on becoming more flexible or resilient by developing their well-being.” How does this arise from the present analysis?

Experimental design

Overall, the connection between aims, analysis, and conclusions could be clarified (see other sections).

Validity of the findings

Reading through, I did not see the “Statistical methods” section requested by the other reviewer. I also did not fully understand the authors’ rebuttal of my comment related to data analysis; namely, why the authors have collected a control group but do not provide any formal statistical comparison between patients and controls with respect to their primary outcome (life satisfaction in Table 2), other than the simple mean difference (Table 1)? Even though the statistical methods are standard, it seems that a short formal “Statistical methods” section would be needed to rigorously map the conducted analyses with the main study questions. As far as I see, the core of the authors’ rebuttal was: “we also investigated the effect of personality and outlook on time on individuals’ level of life satisfaction. This was investigated in order to understand how factors that influence life satisfaction in a clinical population might be targeted in interventions among anorexia patients” (note, the control group was not even mentioned). In the Discussion, however, the authors go on writing “personality did not seem to play the same kind of role for patients as for controls. This is against most studies on life satisfaction, which show that emotional stability…”. But is the difference in regression slopes of patients (0.08) versus controls (0.32) statistically significant? Does the group difference have anything to do with the ”time perception” variables or is it because of the general attenuation of personality-trait slopes (which differed between the groups to begin with) after the addition of the time variables to the models? I do not mean that these are necessarily key tests, but the key tests should preferably be explicitly stated before Results section and then conducted in the Results section.

Additional comments

The manuscript has improved a lot in terms of literature connections, but its internal connections between aims, analysis, and conclusions could be further clarified (see the other sections). As a minor point, please, clarify the discussion on the “risk of overmatching”, considering how representative university students are of the (general?) population from which the patients derived.

---

## Round 0.3 · accepted · Accept

· Academic Editor

Accept

I would like to congratulate the authors for improving the manuscript. It is now acceptable for publication.